# Co-occurrence of the tigecycline resistance gene cluster *tmexCD3-toprJ1* and verona integron-encoded metallo-β-lactamase (VIM) in a human-derived *Pseudomonas sediminis* isolate

Heng Shen,[1] Aoxiao Chen,[2] Chang Cai,[3] Rong Zhang,[2] Hongwei Zhou,[2] Dazhi Jin[1]

**ABSTRACT**   As the last line of defense against carbapenemase-producing Enterobacterales, the growing prevalence of tigecycline resistance poses a serious challenge to clinical therapy. Although *tmexCD-toprJ* clusters were initially described in Enterobacterales, increasing reports in non-fermenting gram-negative bacteria, including *Pseudomonas* species, suggest potential cross-species dissemination (N. Dong, Y. Zeng, Y. Wang, C. Liu, J. Lu, C. Cai, X. Liu, Y. Chen, Y. Wu, Y. Fang, Y. Fu, Y. Hu, H. Zhou, J. Cai, F. Hu, S. Wang, Y. Wang, Y. Wu, G. Chen, Z. Shen, S. Chen, R. Zhang, Lancet Microbe 3:e846–e856, 2022 https://doi.org/10.1016/S2666-5247(22)00221-X). Here, we report a *Pseudomonas sediminis* which exhibited reduced susceptibility to tigecycline (minimum inhibitory concentration > 8 mg/L) carrying *tmexCD3-toprJ1* and $bla_{VIM-2}$ from a child's intestinal tract. Genomic analysis revealed that the gene cluster is present in different hosts and associated with a complex genetic environment.

**IMPORTANCE**   Tigecycline is regarded as one of the last options against multidrug-resistant gram-negative bacteria, especially carbapenemase-producing bacteria. Here, we describe a strain of *Pseudomonas sediminis* (C3-T-78-1) in China carrying tigecycline resistance gene cluster *tmexCD3-toprJ1* and carbapenemase resistance $bla_{VIM-2}$ in a complex gene environment. We highlight that these genes have the potential to disseminate across bacterial hosts and may represent a potential reservoir for antimicrobial resistance dissemination.

**KEYWORDS**   tmexCD3-toprJ1, *Pseudomonas sediminis*, $bla_{VIM-2}$, tigecycline, multidrug-resistant

The emergence of tigecycline resistance genes has posed a significant challenge to the clinical treatment of infections caused by carbapenem-resistant gram-negative bacteria. Since the first report of the plasmid-mediated RND family efflux pump gene cluster *tmexCD1-toprJ1* in humans, animals, and animal products across multiple regions in China in 2020 (1), one of its variants, *tmexCD3-toprJ1b*, was first reported in *Proteus mirabilis* in 2022 (2). Furthermore, clinical cases harboring this gene cluster are extremely rare (0.2%) (1/437) (3), and no studies have yet reported the presence of this gene cluster in *Pseudomonas sediminis*. Therefore, this study provides additional genomic evidence regarding *P. sediminis* carrying the *tmexCD3-toprJ1* gene cluster, represented by *strain* C3-T-78-1 isolated from a child's intestinal tract.

   *P. sediminis* C3-T-78-1 was isolated from the feces of an 8-year-old child with gastrointestinal dysfunction at the *Second Affiliated Hospital of Zhejiang University School of Medicine* in May 2025. The child was treated with cefaclor. Notably, the child had

Address correspondence to Dazhi Jin, jind@hmc.edu.cn.

Heng Shen and Aoxiao Chen contributed equally to this article. The order of co-first authors was determined based on their relative contributions to the study.

The authors declare no conflict of interest.

no prior exposure to tigecycline. A sterile swab was used to pick a feces sample and inoculated it onto a China Blue selective agar plate containing 2 µg/mL tigecycline, followed by overnight incubation at 35°C. MALDI-TOF-MS (Bruker Daltonics, Germany) identified the isolate, and FastANI recorrected it by next-generation sequencing. Tigecycline resistance gene cluster *tmexCD3-toprJ1b* was identified via PCR (4). Antimicrobial susceptibility testing was performed using the Vitek2 system, and the minimum inhibitory concentrations (MICs) were automatically determined by the microdilution method according to the standards set by the Clinical and Laboratory Standards Institute (CLSI), with the exception of tigecycline.

Since no EUCAST clinical breakpoints are available for *Pseudomonas* spp., tigecycline susceptibility was interpreted descriptively based on MIC distribution. Bacterial genomes were extracted using the HiPure Bacteria DNA Kit (Magen, China) and sequenced by the Illumina NovaSeq PE150 platform. Genome assembly was performed with SPAdes v3.15.1 and annotated using RAST (https://rast.nmpdr.org). FastANI was used for species verification, and antimicrobial resistance genes were identified using ABRicate v1.0.1 (https://github.com/tseemann/abricate). A phylogenetic tree was constructed based on core genome alignment using Snippy v4.6.0, visualizing with iTOL v7 (https://itol.embl.de) and the genetic environment surrounding the *tmexCD3-toprJ1* gene cluster was compared using EasyFig v2.2.5.

MALDI-TOF-MS initially identified C3-T-78-1 as *Pseudomonas oleovorans*. However, whole-genome sequencing and FastANI analysis revealed a 95.77% average nucleotide identity (ANI) with the *P. sediminis* reference strain B10D7D (GenBank accession: CP060009.1), supporting its classification as *P. sediminis*.

The isolate met criteria for multidrug-resistant (MDR), exhibiting resistance to ticarcillin/clavulanate, piperacillin/tazobactam, cefoperazone/sulbactam, tobramycin, ciprofloxacin, levofloxacin, and tigecycline. It was categorized as intermediate to ceftazidime and colistin and remained susceptible to imipenem, meropenem, and amikacin (Table 1).

The tigecycline-resistant gene cluster *tmexCD-toprJ* exhibited 100% amino acid similarity to *tmexCD3-toprJ1* when aligned with the NCBI database. In addition, the strain harbors multiple resistance determinants, including the carbapenem resistance gene $bla_{VIM-2}$, the aminoglycoside resistance gene *aacA4*, the sulfonamide resistance gene *sul1*, the quaternary ammonium compound resistance gene *qacEdelta1*, and the mercury resistance genes (*merEDAPTR*). Genomic localization predicted using the KleTY database (5) indicated that these genes are chromosomally encoded.

**TABLE 1** MIC results for C3-T-78-1[a]

| Antibiotics | C3-T-78-1 | |
| --- | --- | --- |
| | MIC | Interpretation |
| TIC | ≥128 | R |
| TZP | ≥128 | R |
| CAZ | 16 | I |
| SCF | ≥64 | R |
| FEP | 4 | I |
| IMP | 2 | S |
| MEM | 1 | S |
| AK | ≤2 | S |
| TOB | 8 | R |
| CIP | 2 | R |
| LVX | 4 | R |
| TGC | ≥8 | R |
| PE | 2 | I |

[a]* MIC, minimal inhibitory concentration, units are mg/L. "R", "S", "I" represents resistant, sensitive and intermediate respectively. TIC, ticarcillin/clavulanate; TZP, piperacillin/tazobactam; CAZ, ceftazidime; SCF, cefoperazone/sulbactam; FEP, cefepime; IMP, imipenem; MEM, meropenem; AK, amikacin; TOB, tobramycin; CIP, ciprofloxacin;LVX, levofloxacin; TGC, tigecycline; PE, colistin.

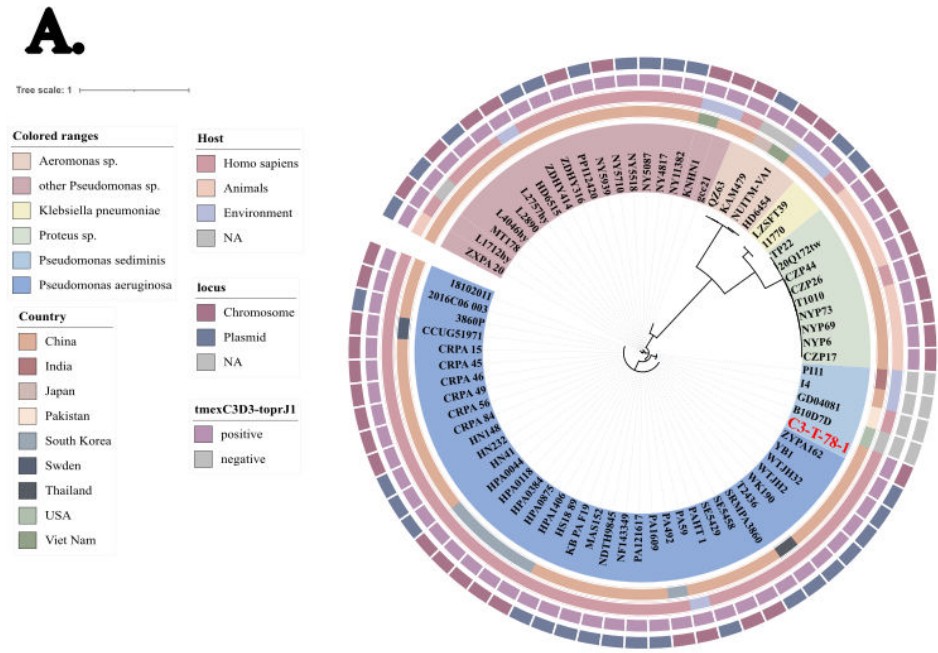

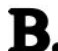

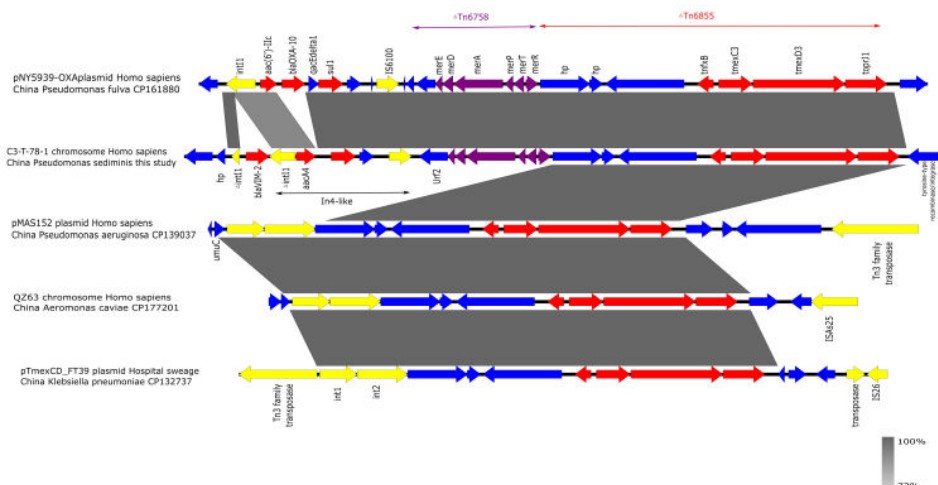

FIG 1 (A) Phylogenetic tree based on core genome SNPs of 75 strains carrying or lacking *tmexCD3-toprJ1* gene cluster. The different background colors of the sequence ID represented the genus of bacteria. (B) the genetic environment of the *tmexCD3-toprJ1* gene cluster among different strains. Regions with > 99% similarity are shaded in grey. blue arrows represent genes with hypothetical protein or other functions; yellow arrows show insertion or integration functions; Red arrows indicate antibiotic resistance genes; purple arrows represent mercury resistance genes.

The *tmexCD3-toprJ1* gene cluster has also been identified in clinical isolates from various regions in China, such as *Pseudomonas aeruginosa* MAS152 plasmid (pMAS152, GenBank accession number CP139037), *Aeromonas caviae* QZ63 chromosome (GenBank accession number CP177201), and the hospital wastewater isolate *Klebsiella pneumoniae* LZSFT39 plasmid pTmex_FT39 (GenBank accession number CP132737), suggesting that this cluster can be mobilized between plasmids and chromosomes across different bacterial genera. Unlike the clusters found in these strains, which are embedded within

Tn6855, the *tmexCD3-toprJ1* cluster in C3-T-78-1 is associated with a complex genetic environment (Fig. 1A). Specifically, $bla_{VIM-2}$ is inserted into intI1, with an *intI1-aacA4-qacE delta1-sul1*-IS*6100* structure forming an In4-like class 1 integron (GenBank accession number GQ293499). Interestingly, C3-T-78-1 carries carbapenem-resistant gene $bla_{VIM-2}$, but it shows sensitivity to imipenem and meropenem. Previous studies have shown that $bla_{VIM-2}$ acts as a resistance gene cassette within class 1 integrons and lacks its own promoter, relying instead on the promoter for cassettes (Pc) for expression (6). Analysis of the genomic sequence of the strain C3-T-78-1 revealed that $bla_{VIM-2}$ was inserted within the integrase gene *intI1*, rather than within the resistance gene cassette of the class 1 integron, which may explain why the strain, despite carrying the $bla_{VIM-2}$ gene, exhibited a carbapenem-susceptible phenotype. The *urf2-merE-merD-merA-merP-merT-merR* structure forms ΔTn6758 (GenBank accession number JX448550) and *hp-hp-hp-tnxfB-tmexCD3-toprJ1* corresponds to ΔTn6855 (GenBank accession number MK347425). This genetic environment shows high similarity to a segment of *Pseudomonas fulva* NY5939 plasmid pNY5939-OXA, isolated from clinical bile samples, sharing 99.99% nucleotide identity over 89% of the sequence (GenBank accession number CP161880). Previous studies indicate that *tmexCD-toprJ* clusters originate from the chromosomes of *Pseudomonas* or *Aeromonas* genus (1, 7), and the surrounding mobile elements suggest that they may subsequently mobilize onto plasmids via integrase-mediated events, thereby horizontal gene transfer within or between species (8, 9).

Phylogenetic analysis was performed on 70 *tmexCD3-toprJ1*-positive strains, four *tmexCD3-toprJ1*-negative *P. sediminis* strains from the NCBI database, and strain C3-T-78-1 to investigate the distribution of the *tmexCD3-toprJ1* gene cluster in *Pseudomonas* and other genera (Fig. 1B). Among these, a total of 58 clinical isolates harboring the *tmexCD3-toprJ1* gene cluster were identified, originating from sputum, bronchoalveolar lavage fluid, or other respiratory specimens (*n* = 12), urine (*n* = 11), feces (*n* = 4), bile (*n* = 4), blood (*n* = 3), and swabs (*n* = 3). In addition, 12 isolates were recovered from animals, mainly from fecal samples, and 11 environmental isolates, including 3 from hospital sewage and 5 from wastewater. The cluster was chromosomally encoded in 48 isolated (64%), 37 isolates (49.3%) carried it on plasmids; several strains harbored both chromosomal and plasmid copies. Although *P. sediminis* is phylogenetically close to *P. aeruginosa*, the genetic environment of the *tmexCD3-toprJ1* cluster in C3-T-78-1 differs from that in *P. aeruginosa* MAS152 and instead shows greater similarity in mobile genetic elements to *P. fulva* NY5939.

In conclusion, we observed a multidrug-resistant *P. sediminis containing* tigecycline and carbapenem resistance gene, while the failure of $bla_{VIM-2}$ expression may be related to its insertion site within the integron. To our knowledge, this is the first time that the genetic environment of the *tmexCD3-toprJ1* gene cluster in human-derived *P. sediminis* is studied worldwide. Although tigecycline is not approved for veterinary use, doxycycline and other antibiotics remain widely applied in livestock feed (10). Under such selective pressure, and facilitated by mobile genetic elements and integrase-mediated recombination, this gene cluster has the potential to disseminate across bacterial hosts, highlighting the importance of surveillance.

## ACKNOWLEDGMENTS

We sincerely thank all participants for their contributions to this study.

Conceptualization: Chang Cai, Methodology: Hongwei Zhou, Data Curation: Rong Zhang, Formal Analysis: Heng Shen, Aoxiao Chen Investigation: Rong Zhang, Hongwei ZhouWriting – Original Draft Preparation: Heng Shen, Aoxiao Chen Writing – Review & Editing: Chang Cai, Rong Zhang Visualization: Heng Shen, Aoxiao Chen Supervision: Dazhi Jin Project Administration: Rong Zhang, Dazhi JinAll authors have read and agreed to the final version of the manuscript.

## AUTHOR AFFILIATIONS

[1]Hangzhou Medical College, Hangzhou, Zhejiang, China

[2]Clinical Microbiology Laboratory, The Second Affiliated Hospital of Zhejiang University School of Medicine, Zhejiang University, Hangzhou, China

[3]Sanya Institute of Nanjing Agricultural University, Laboratory of Emerging Infectious Diseases and One Health, College of Veterinary Medicine, Nanjing Agricultural University, Nanjing, China

## AUTHOR ORCIDs

Heng Shen  http://orcid.org/0009-0000-0641-4552
Aoxiao Chen  http://orcid.org/0009-0004-4797-9524
Rong Zhang  http://orcid.org/0000-0002-2174-7985
Hongwei Zhou  http://orcid.org/0000-0002-9478-4023
Dazhi Jin  http://orcid.org/0000-0002-2613-7844

## AUTHOR CONTRIBUTIONS

Heng Shen, Writing – original draft | Aoxiao Chen, Writing – original draft | Chang Cai, Conceptualization | Rong Zhang, Resources | Hongwei Zhou, Methodology.

## DATA AVAILABILITY

The genome assemblies of C3-T-78-1 have been deposited in the National Center for Biotechnology Information (NCBI) and are registered under BioProject accession number PRJNA1429045.

## ADDITIONAL FILES

The following material is available online.

### Open Peer Review

**PEER REVIEW HISTORY (review-history.pdf).** An accounting of the reviewer comments and feedback.

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
