## [Reviewer comments · Microbiology Spectrum]

Microbiology Spectrum

Co-occurrence of the tigecycline resistance gene cluster *tmexCD3-toprJ1* and verona integron-encoded metallo- β -lactamase (VIM) in a human-derived *Pseudomonas sediminis* isolate

Heng Shen, Aoxiao Chen, Chang Cai, Rong Zhang, Hongwei Zhou, and Dazhi Jin

Corresponding Author(s): Dazhi Jin, Hangzhou Medical College

Review Timeline:

Submission Date:	October 31, 2025
Editorial Decision:	November 17, 2025
Revision Received:	November 19, 2025
Editorial Decision:	February 11, 2026
Revision Received:	March 2, 2026
Accepted:	March 4, 2026

Editor: Gabriele Arcari

Reviewer(s): The reviewers have opted to remain anonymous.

Transaction Report:

DOI: <https://doi.org/10.1128/spectrum.03491-25>

Re: Spectrum03491-25 (Co-occurrence of tigeicycline-resistant gene cluster tmexCD3-toprJ1 and verona integron-encoded metallo- β -lactamase (VIM) in *Pseudomonas sediminis*)

Dear Dr. Dazhi Jin:

I am pleased to inform you that your manuscript has been editorially accepted for publication. However, there are a few additional questions in the submission form that need to be answered before the final decision. Once these are completed, please return your submission so that I can move your paper forward to acceptance.

Below you will find my comments and instructions from the Spectrum editorial office.

Revision Guidelines

Sincerely,
Gabriele Arcari
Editor
Microbiology Spectrum

Dear Editor:

Thank you for your valuable comments. We have carefully considered each comment and made revisions accordingly. Below is our response to your comments.

Editor comments: I am pleased to inform you that your manuscript has been editorially accepted for publication. However, there are a few additional questions in the submission form that need to be answered before the final decision. Once these are completed, please return your submission so that I can move your paper forward to acceptance. Below you will find my comments and instructions from the Spectrum editorial office.

Note the following requirements:

- Upload point-by-point responses to the issues raised by the reviewers in a file named "Response to Reviewers," NOT in your cover letter.
- Upload a compare copy of the manuscript (without figures) as a "Marked-Up Manuscript" file.
- Upload a clean .DOC/.DOCX version of the revised manuscript and remove the previous version.
- Each figure must be uploaded as a separate, editable, high-resolution file (TIFF or EPS preferred), and any multipanel figures must be assembled into one file.
- Any supplemental material intended for posting by ASM should be uploaded with their legends separate from the main manuscript. You can combine all supplemental material into one file (preferred) or split it into a maximum of 10 files with all associated legends included.

Response: Thank you for your valuable comments. We have revised the manuscript according to the revision guidelines provided on the journal's webpage.

Re: Spectrum03491-25R1 (Co-occurrence of tigeicycline-resistant gene cluster tmexCD3-toprJ1 and verona integron-encoded metallo- β -lactamase (VIM) in *Pseudomonas sediminis*)

Dear Dr. Dazhi Jin:

Thank you for the privilege of reviewing your work. Below you will find my comments, instructions from the Spectrum editorial office, and the reviewer comments.

Revision Guidelines

Sincerely,
Gabriele Arcari
Editor
Microbiology Spectrum

Reviewer #2 (Comments for the Author):

Overall, the study is well-structured and systematically presented. The work contributes meaningfully to tracing the potential origin and genetic context of the tmexCD3-toprJ cluster. However, several comments should be addressed to further strengthen the manuscript, as detailed below:

1. In the abstract, the authors first describe the challenge of tigecycline resistance in Enterobacteriales. However, as the study primarily focuses on *Pseudomonas sediminis*, it is suggested to briefly address whether the *tmexCD-toprJ* cluster carried in Enterobacteriales shows any relevant association with *Pseudomonas* species.
2. After consulting the current European Committee on Antimicrobial Susceptibility Testing (EUCAST) guidelines (<https://www.eucast.org/bacteria/clinical-breakpoints-and-interpretation/clinical-breakpoint-tables/>), I did not find interpretive criteria for tigecycline applicable to *Pseudomonas* spp. Thus, the description "tigecycline-resistant *Pseudomonas sediminis*" requires more precise scientific wording. If other editions or years of the EUCAST guidelines were consulted, please provide the corresponding version information.
3. If the complete genome sequence of strain C3-T-78-1 has been deposited in a public database, please provide the corresponding accession number.
4. The strain C3-T-78-1, though carrying VIM-2, remains susceptible to carbapenems. The authors suggest that this observation may be attributed to the gene not being located within an expressible resistance cassette in the integron. Furthermore, published studies indicate that genomic rearrangements in carbapenemase-positive isolates can also lead to susceptible phenotypes in routine antimicrobial susceptibility testing (doi: 10.1016/j.drug.2024.101123./ doi: 10.1093/jac/dkt235.). It is recommended that the authors discuss this potential mechanism in the context of their findings.
5. Please ensure consistent spacing between text and parentheses throughout the manuscript.
6. "Enterbacteriale": Please check the spelling.

Reviewer #3 (Comments for the Author):

This manuscript from Heng Shen and colleagues describes a multidrug-resistant *Pseudomonas sediminis* isolate carrying the tigecycline resistance gene cluster *tmexCD3-toprJ1* together with *blaVIM-2*.

The work features a well documented genetic environment and comparative context, within depth comparative analyses

Nonetheless, the clinical significance of this single colonizing isolate is somewhat overstated, the interpretation of the *blaVIM-2* finding is speculative, and the English language and style require careful editing.

Major comments

While the isolate was obtained from the feces of a child with gastrointestinal dysfunction who received cefaclor and had no prior exposure to tigecycline, there is no evidence of invasive infection or treatment failure.

Hence it is plausible that what Authors describe is an intestinal colonization case, yet the Abstract, Importance, and concluding sentences repeatedly frame the finding as posing a "serious challenge to clinical therapy" and as "raising serious concerns for public health." For a single colonizing *P. sediminis* isolate, this language is too strong.

This is true as well in the conclusion, where Authors may want to emphasize the potential for spread and the surveillance relevance, rather than immediate clinical impact.

The manuscript currently states that this is "the first time that the genetic environment of the *tmexCD3-toprJ1* gene cluster in human-derived *Pseudomonas sediminis* was reported all over the world." Absolute "first worldwide" statements are difficult to sustain given the quickly expanding literature on *tmexCD-toprJ*, and this is even more true given fact that *P. sediminis* is a not-so-well characterized species.

The finding that the isolate carries *blaVIM-2* yet remains susceptible to imipenem and meropenem is one of the most interesting aspects of the report. However, the explanation offered ("*blaVIM-2* is not located within an expressible antibiotic resistance gene cassette of the integron, so it can not produce functional VIM") is speculative and not supported by functional data. To state the Authors should assess the presence of a recognizable promoter upstream of the gene.

If Authors do not wish do carry out additional experiments (e.g., phenotypic test or a quantitative analysis of *blaVIM-2* expression) I suggest softening the wording.

Minor comments

A careful language edit is needed

In figure 1, panel A explicitly labelling which sequences are chromosomal vs plasmid for all examples, not only in the text would help interpretation

Reviewer: 2

Comments to the Author

Overall, the study is well-structured and systematically presented. The work contributes meaningfully to tracing the potential origin and genetic context of the *tmexCD-toprJ* cluster. However, several comments should be addressed to further strengthen the manuscript, as detailed below:

1. In the abstract, the authors first describe the challenge of tigecycline resistance in Enterobacteriales. However, as the study primarily focuses on *Pseudomonas sediminis*, it is suggested to briefly address whether the *tmexCD-toprJ* cluster carried in Enterobacteriales shows any relevant association with *Pseudomonas* species.

Response: We thank the reviewer for this helpful suggestion. We agree that a clearer transition was necessary. We have revised the abstract to clarify that *tmexCD-toprJ* clusters were initially described in *Enterobacteriales* but have increasingly been reported in non-fermenting Gram-negative bacteria, including *Pseudomonas spp.*, suggesting possible cross-species dissemination. The revised sentence has been incorporated in the Abstract (Lines 18–21).

2. After consulting the current European Committee on Antimicrobial Susceptibility Testing (EUCAST) guidelines (<https://www.eucast.org/bacteria/clinical-breakpoints-and-interpretation/clinical-breakpoint-tables/>), I did not find interpretive criteria for tigecycline applicable to *Pseudomonas spp.* Thus, the description "tigecycline-resistant *Pseudomonas sediminis*" requires more precise scientific wording. If other editions or years of the EUCAST guidelines were consulted, please provide the corresponding version information.

Response: We appreciate this important technical correction. We have revised the

manuscript to avoid using the term "tigecycline-resistant" for *Pseudomonas sediminis*. Since no EUCAST clinical breakpoint is available for *Pseudomonas spp.*, tigecycline susceptibility is now described based on MIC values. The revised wording now reads: "Pseudomonas sediminis which exhibited reduced susceptibility to tigecycline (MIC > 8 mg/L)." These revisions have been made in the Methods and Results sections (Lines 22-23 and Lines 62–64).

3. If the complete genome sequence of strain C3-T-78-1 has been deposited in a public database, please provide the corresponding accession number.

Response: Thank you for pointing this issue out. We have provided the BioProject accession Number in the manuscript.

4. The strain C3-T-78-1, though carrying VIM-2, remains susceptible to carbapenems. The authors suggest that this observation may be attributed to the gene not being located within an expressible resistance cassette in the integron. Furthermore, published studies indicate that genomic rearrangements in carbapenemase-positive isolates can also lead to susceptible phenotypes in routine antimicrobial susceptibility testing (doi: 10.1016/j.drug.2024.101123./ doi: 10.1093/jac/dkt235.). It is recommended that the authors discuss this potential mechanism in the context of their findings.

Response: We thank the reviewer for this valuable suggestion. Previous studies have shown (doi: 10.1128/AAC.44.4.891-897.2000) that *bla*_{VIM-2} acts as a resistance gene cassette within class 1 integrons and lacks its own promoter, relying instead on the promoter for cassettes (P_c) for expression. Analysis of the genomic sequence of The strain C3-T-78-1, revealed that *bla*_{VIM-2} was inserted within the integrase gene *intI1*, rather than within the resistance gene cassette of the class 1 integron, which explains why the strain, despite carrying the *bla*_{VIM-2} gene, exhibited a carbapenem-susceptible phenotype (Lines 108–115).

5. Please ensure consistent spacing between text and parentheses throughout the manuscript.

Response: We have carefully revised the manuscript to ensure consistent formatting and spacing throughout.

6. "Enterbacteriale": Please check the spelling.

Response: The spelling has been corrected to "Enterobacterales" (Lines 17).

Reviewer: 3

Comments to the Author

This manuscript from Heng Shen and colleagues describes a multidrug-resistant *Pseudomonas sediminis* isolate carrying the tigecycline resistance gene cluster *tmexCD3-toprJ1* together with *bla*_{VIM-2}.

The work features a well documented genetic environment and comparative context, within depth comparative analyses

Nonetheless, the clinical significance of this single colonizing isolate is somewhat overstated, the interpretation of the *bla*_{VIM-2} finding is speculative, and the English language and style require careful editing.

Response: We sincerely thank the reviewer for the thorough evaluation and insightful comments. We have carefully revised the manuscript to address all concerns raised.

Major comments

1. While the isolate was obtained from the feces of a child with gastrointestinal dysfunction who received cefaclor and had no prior exposure to tigecycline, there is no evidence of invasive infection or treatment failure.

Hence it is plausible that what Authors describe is an intestinal colonization case, yet the Abstract, Importance, and concluding sentences repeatedly frame the finding as posing a "serious challenge to clinical therapy" and as "raising serious concerns for

public health." For a single colonizing *P. sediminis* isolate, this language is too strong. This is true as well in the conclusion, where Authors may want to emphasize the potential for spread and the surveillance relevance, rather than immediate clinical impact.

Response: We thank the reviewer for this important perspective. We agree that the clinical implications of a single colonization isolate should not be overstated. Accordingly, we have revised the Abstract, Importance section, Discussion, and Conclusion to soften statements implying direct clinical threat. Phrases such as "this study provides additional genomic evidence regarding" and "raises serious public health concern" have been replaced with more balanced wording (Lines 44–45 and Lines 154-155).

2. The manuscript currently states that this is "the first time that the genetic environment of the *tmexCD3-toprJ1* gene cluster in human-derived *Pseudomonas sediminis* was reported all over the world." Absolute "first worldwide" statements are difficult to sustain given the quickly expanding literature on *tmexCD-toprJ*, and this is even more true given fact that *P. sediminis* is a not-so-well characterized species.

Response: We appreciate this important correction. We have removed the statement claiming a "global first report". In Lines 147–148, our original sentence reads "To our knowledge, this is the first report describing the genetic environment of the *tmexCD3-toprJ1* gene cluster in human-derived *Pseudomonas sediminis* worldwide", We believe that the wording is appropriately cautious and consistent with the reviewer's suggestion. Therefore, we respectfully prefer to retain the original phrasing at this stage.

3. The finding that the isolate carries *bla*_{VIM-2} yet remains susceptible to imipenem and meropenem is one of the most interesting aspects of the report. However, the explanation offered ("*bla*_{VIM-2} is not located within an expressible antibiotic resistance

gene cassette of the integron, so it can not produce functional VIM") is speculative and not supported by functional data. To state the Authors should assess the presence of a recognizable promoter upstream of the gene.

If Authors do not wish do carry out additional experiments (e.g., phenotypic test or a quantitative analysis of *bla*_{VIM-2} expression) I suggest softening the wording.

Response: We thank the reviewer for this thoughtful comment. We agree that our original wording was overly definitive.

We did not identify any recognizable promoter sequences upstream of the *bla*_{VIM-2} gene in the genome. Previous studies (DOI: 10.1128/AAC.44.4.891-897.2000) have demonstrated that *bla*_{VIM-2}, as a resistance gene cassette within class 1 integrons, relies on the promoter for cassettes (Pc) for expression. In the present study, *bla*_{VIM-2} was inserted within the integrase gene *intI1* rather than in the resistance gene cassette region of a class 1 integron, which may account for the carbapenem-susceptible phenotype despite carriage of the *bla*_{VIM-2} gene. (Lines 108–115)

Minor comments

1. A careful language edit is needed

Response: The manuscript has undergone comprehensive language editing to improve clarity, grammar, and scientific tone.

2. In figure 1, panel A explicitly labelling which sequences are chromosomal vs plasmid for all examples, not only in the text would help interpretation

Response: The chromosomal or plasmid source of each sequence has been indicated directly after the corresponding strain number on the left side of the figure.

Final Statement:

We sincerely thank both reviewers for their insightful comments, which have substantially improved the clarity, rigor, and balance of our manuscript. We believe that the revised version addresses all concerns and meets the journal's standards.

Re: Spectrum03491-25R2 (Co-occurrence of the tigeicycline resistance gene cluster *tmexCD3-toprJ1* and verona integron-encoded metallo- β -lactamase (VIM) in a human-derived *Pseudomonas sediminis* isolate)

Dear Dr. Dazhi Jin:

Your manuscript has been accepted, and I am forwarding it to the ASM production staff for publication. Your paper will first be checked to make sure all elements meet the technical requirements. ASM staff will contact you if anything needs to be revised before copyediting and production can begin. Otherwise, you will be notified when your proofs are ready to be viewed.

Sincerely,
Gabriele Arcari
Editor
Microbiology Spectrum